# Association between Visceral Adiposity Index, Binge Eating Behavior, and Grey Matter Density in Caudal Anterior Cingulate Cortex in Severe Obesity

**DOI:** 10.3390/brainsci11091158

**Published:** 2021-08-31

**Authors:** Sylvain Iceta, Mahsa Dadar, Justine Daoust, Anais Scovronec, Vicky Leblanc, Melissa Pelletier, Laurent Biertho, André Tchernof, Catherine Bégin, Andreanne Michaud

**Affiliations:** 1Research Center of the Quebec Heart and Lung Institute, Université Laval, Quebec City, QC G1V 4G5, Canada; justine.daoust.1@ulaval.ca (J.D.); anais.scovronec.1@ulaval.ca (A.S.); melissa.pelletier@criucpq.ulaval.ca (M.P.); andre.tchernof@criucpq.ulaval.ca (A.T.); catherine.begin@psy.ulaval.ca (C.B.); 2School of Nutrition, Université Laval, Quebec City, QC G1V 0A6, Canada; 3CERVO Brain Research Center, Centre Intégré Universitaire Santé et Services Sociaux de la Capitale Nationale, Université Laval, Quebec City, QC G1E 1T2, Canada; mahsa.dadar.1@ulaval.ca; 4Centre Nutrition, Santé et Société (NUTRISS), Institut sur la Nutrition et les Aliments Fonctionnels (INAF), Université Laval, Quebec City, QC G1V 0A6, Canada; vicky.leblanc@fsaa.ulaval.ca; 5Département de Chirurgie Générale, Quebec Heart and Lung Institute, Université Laval, Quebec City, QC G1V 4G5, Canada; laurentbiertho@gmail.com; 6School of Psychology, Université Laval, Quebec City, QC G1V 0A6, Canada

**Keywords:** binge eating behavior, grey matter density, severe obesity, metabolic disorders, visceral adiposity, emotion regulation

## Abstract

Visceral adipose tissue accumulation is an important determinant of metabolic risk and can be estimated by the visceral adiposity index (VAI). Visceral adiposity may impact brain regions involved in eating behavior. We aimed to examine the association between adiposity measurements, binge eating behavior, and grey matter density. In 20 men and 59 women with severe obesity, Grey matter density was measured by voxel-based morphometry for six regions of interest associated with reward, emotion, or self-regulation: insula, orbitofrontal cortex, caudal and rostral anterior cingulate cortex (ACC), ventromedial prefrontal cortex (vmPFC), and dorsolateral prefrontal cortex (DLPFC). Binge eating behavior, depression and impulsivity was assessed by the Binge Eating Scale, Beck Depression Inventory and UPPS Impulsive Behavior Scale, respectively. Men and women were distinctively divided into two subgroups (low-VAI and high-VAI) based on the mean VAI score. Women with high-VAI were characterized by metabolic alterations, higher binge eating score and lower grey matter density in the caudal ACC compared to women with low-VAI. Men with high-VAI were characterized by a higher score for the sensation-seeking subscale of the UPPS–Impulsive Behavior Scale compared to men with low-VAI. Using a moderation–mediation analysis, we found that grey matter density in the caudal ACC mediates the association between VAI and binge eating score. In conclusion, visceral adiposity is associated with higher binge eating severity in women. Decreased grey matter density in the caudal ACC, a region involved in cognition and emotion regulation, may influence this relationship.

## 1. Introduction

Excess body fat, especially within the abdominal cavity, is strongly related to metabolic alterations such as insulin resistance, type 2 diabetes, dyslipidemia, and low-grade chronic inflammation [1,2,3]. Considering that body mass index (BMI) alone cannot identify individuals with excess visceral adiposity and that waist circumference cannot distinguish subcutaneous from visceral adiposity, various simple anthropometric tools have been developed over the past years to predict excess visceral adiposity and identify individuals at high risk for cardiometabolic abnormalities [3]. The visceral adiposity index (VAI), a sex-specific mathematical index based on BMI, waist circumference, triglycerides, and high-density lipoprotein cholesterol levels is one of these tools [4,5,6]. 

There is accumulating evidence that a subset of individuals with obesity, especially severe obesity, are at increased risk of disordered eating such as binge eating [7,8]. Binge eating is a behavior characterized by the consumption of larger-than-normal amounts of food in a short period of time, with feelings of distress in the absence of regular compensatory behaviors [9]. Binge eating is also associated with a lack of control over eating and constitutes the last stage of the uncontrolled eating spectrum suggested by Vainik et al. [10]. According to the DSM-5, binge eating disorder (BED) is characterized by recurrent episodes of binge eating that occur at least once per week for three months and its prevalence is higher in women compared to men [11]. Cross-sectional and longitudinal evidence also suggests that binge eating is associated with cardiometabolic alterations, including insulin resistance, hypertension, dyslipidemia, cancer and type 2 diabetes [12,13,14,15,16,17,18]. Cardiometabolic alterations may increase the risk of mortality and may reduce longevity [3,19]. The increased cardiometabolic risk associated with disrupted eating behavior can be attributed to excess visceral adiposity [20,21,22]. However, little is known regarding the association between visceral adiposity and binge eating behavior, especially in individuals with severe obesity. 

Many theoretical frameworks have been proposed to understand the mechanisms underlying obesity and binge eating behavior. Impulsivity, which is considered as a tendency to act rashly without full consideration of the consequences, has been proposed as a core component of binge eating [23]. The neurobehavioral processes that lead to impulsivity result from the interaction of high reward sensitivity, low self-control, and high negative effect [23,24,25,26]. These psychological constructs are also recognized as risk factors for obesity [10,26,27] and are related to different brain systems: (i) the limbic system involved in rewarding processes, which includes the striatum, ventromedial prefrontal cortex (vmPFC), the orbitofrontal cortex (OFC), the insula, and the hippocampus; (ii) the frontoparietal networks (mostly the dorsolateral prefrontal cortex (DLPFC) and the anterior cingulate cortex (ACC)), involved in self-regulation and cognitive control processes; and (iii) the hypothalamic-pituitary-adrenal (HPA) axis, involved in stress responses [10,26]. 

Recent studies using voxel-based morphometry provided evidence that individuals with obesity are characterized by grey matter atrophy in brain regions involved in reward, self-regulation and emotional regulation processes, namely the OFC, vmPFC, cerebellum, and areas in the temporal and parietal poles [28,29,30,31]. A recent study also reported that individuals with severe obesity have reduced grey matter density in frontal, parietal and temporal regions compared to individuals with normal weight [32]. Voon et al. also reported lower grey matter volume in ventral striatal, OFC, and caudate in individuals characterized by obesity and BED compared to those without BED, thus highlighting the unique contribution of binge eating [33]. However, most of the studies examining structural brain changes used BMI, which is a limited tool to evaluate the risk of obesity-associated metabolic alterations [4,6,34,35,36,37,38]. It has been suggested that metabolic alterations associated with visceral fat accumulation, including insulin resistance, can impact brain regions involved in eating behavior [39,40]. Thus, a better understanding of the link between visceral adiposity, binge eating behavior, and brain structure in regions involved in reward, self-regulatory, and negative emotionality processes is of particular importance [41]. Considering that there are large-scale studies showing sex-differences in brain morphology, body fat distribution and eating behaviors [3,42,43,44], it is of particular importance to examine this link in men and women separately.

In the current study, we first aimed: (i) to examine the association between adiposity measurements and binge eating behavior; and (ii) to compare binge eating behavior severity, impulsivity level and grey matter density in regions of interest (ROIs) (insula, OFC, caudal, and rostral ACC, vmPFC, DLPFC) in women and men presenting severe obesity with high- versus low-visceral adiposity, based on the VAI index. We hypothesized that adiposity is positively associated with binge eating severity and that participants with higher VAI score have higher binge eating severity and lower grey matter density in brain regions involved in reward sensitivity, self-control, and emotional regulation, especially in women. Following these analyses, we then explored whether the association between visceral adiposity and binge eating severity is mediated by changes in grey matter density of our ROIs. 

## 2. Materials and Methods

This study is part of a larger study on the effects of restrictive and malabsorptive bariatric procedures on obesity-associated metabolic impairment. For the current study, participants scheduled to undergo bariatric surgery at the Institut universitaire de cardiologie et de pneumologie de Québec (IUCPQ) were recruited. Exclusion criteria for this study were: BMI < 35 kg/m^2^; age < 18 or >60 years; any uncontrolled medical, surgical, neurological, or psychiatric condition; cirrhosis or albumin deficiency; any medication that can affect the central nervous system; pregnancy; substance or alcohol abuse; previous gastric, esophageal, brain or bariatric surgery; gastro-intestinal inflammatory diseases or gastro-intestinal ulcers; severe food allergy; contraindications to MRI (implanted medical device, metal fragment in body, or claustrophobia). Participants were studied approximately 1 month prior to as well as 4-, 12- and 24-months post-surgery. At each visit, participants underwent physical examination, fasting blood biochemistry, anthropometric measurements, bioimpedance analysis, psychological assessment, and an MRI session. For this study, only data prior to surgery were used. The study sample included 79 participants with severe obesity (20 men and 59 women) with a baseline visit between September 2016 and March 2020. All participants provided written informed consent to participate in the study in accordance with the Declaration of Helsinki, and the protocol received approval from the Research Ethics Committee of the Centre de recherche de l’IUCPQ (approval number 2016-2569, 21237). 

### 2.1. Plasma Lipid Profile and Glucose Homeostasis Markers

Blood samples were collected on the morning after a 12 h fast in EDTA-coated tubes or serum clot activator tubes and blood biochemistry were analyzed at the IUCPQ laboratory. Plasma levels of cholesterol, high-density lipoproteins, low-density lipoproteins, triglycerides, Apolipoprotein B, glucose, insulin, HbA1c, and TSH were measured. HOMA-IR index, a marker of hepatic insulin resistance, was calculated with the following formula: (Insulin (pmol/L) × Glucose (mmol/L))/(22.5 × 6).

### 2.2. Anthropometric Measurements

All anthropometric measurements were conducted by trained health professionals following standardized procedures. Height was measured using a stadiometer. Weight and body composition including fat mass and body fat percent were obtained using a calibrated bioelectrical impedance scale (InBody520, body composition analyzer, Biospace, Los Angeles, CA, USA or Tanita DC-430U, Arlington Heights, IL, USA). Hip, waist, and neck circumferences were measured in centimeters to the nearest millimeter using a standardized procedure.

### 2.3. Adiposity Measurements

The following adiposity markers were calculated with the following formulas: Body Mass Index (BMI):
$$\mathrm{BMI}\;\left(\mathrm{kg}\cdot m^{-2}\right)=\frac{\mathrm{Weight}\;\left(\mathrm{kg}\right)}{\mathrm{Height}\;{\left(m\right)}^{2}}$$Waist-to-hip ratio (WHR):
$$\mathrm{WHR}=\frac{\mathrm{Waist}\;\mathrm{circumference}\;\left(\mathrm{cm}\right)}{\mathrm{Hip}\;\mathrm{circumference}\;\left(\mathrm{cm}\right)}$$Percentage of fat mass (%FM):
$$\%\mathrm{FM}=\frac{\mathrm{Fat}\;\mathrm{mass}\;\left(\mathrm{kg}\right)}{\mathrm{Weight}\;\left(\mathrm{kg}\right)}\;\times \;100$$Body fat mass index (BFMI):
$$\mathrm{BFMI}\;\left(\mathrm{kg}\cdot m^{-2}\right)=\frac{\mathrm{Fat}\;\mathrm{mass}\;\left(\mathrm{kg}\right)}{\mathrm{Height}\;{\left(m\right)}^{2}}$$Visceral Adiposity Index (VAI) [5]:

For women:$$V_{\mathrm{isceral}}A_{\mathrm{diposity}}I_{\mathrm{ndex}}=\left(\frac{\mathrm{Waist}\;\mathrm{circumference}\;\left(\mathrm{cm}\right)}{36.58+\left(1.89\times \mathrm{BMI}\;\left(\mathrm{kg}\cdot m^{-2}\right)\right)}\right)\times \frac{\mathrm{TGs}\;\left(\mathrm{mmol}\cdot L^{-1}\right)}{0.81}\times \frac{1.52}{\mathrm{HDLc}\;\left(\mathrm{mmol}\cdot L^{-1}\right)}$$

For men: $$V_{\mathrm{isceral}}A_{\mathrm{diposity}}I_{\mathrm{ndex}}=\left(\frac{\mathrm{Waist}\;\mathrm{circumference}\;\left(\mathrm{cm}\right)}{39.68+\left(1.88\times \mathrm{BMI}\;\left(\mathrm{kg}\cdot m^{-2}\right)\right)}\right)\times \frac{\mathrm{TGs}\;\left(\mathrm{mmol}\cdot L^{-1}\right)}{1.03}\times \frac{1.31}{\mathrm{HDLc}\;\left(\mathrm{mmol}\cdot L^{-1}\right)}$$

### 2.4. Psychological Assessment

Depression, impulsivity, and binge eating behaviors were assessed by the following self-administered questionnaires.
Beck Depression Inventory II (BDI-II)

Current depressive symptoms were assessed using the 21-item version of the Beck Depression Inventory [45,46,47]. The participant was asked to report how he or she had felt during the previous 2 weeks. The scale content reflects the cognitive, affective, somatic, and vegetative symptoms of depression and includes items such as: “*I am so sad or unhappy that I can’t stand it*”, “*I feel my future is hopeless and will only get worse*”, “*I dislike myself*”. Each item was measured on a four-point Likert scale ranging from 0 (equivalent to absence or “as usual”) to 3 (maximum symptoms). The total score ranges between 0 and 63 and are categorized as minimally (0 to 13), mildly (14 to 19), moderately (20 to 28), and severely (29 to 63) depressed. In the current study, overall Cronbach’s alpha was 0.89.
UPPS Impulsive Behavior Scale (UPPS)

Four dimensions of impulsivity were assessed using the 44-item scale UPPS Impulsive Behavior Scale [48,49]: urgency (e.g., “*It is hard for me to resist acting on my feelings*”), lack of premeditation (e.g., “*My thinking is usually careful and purposeful*”), lack of perseverance (e.g., “*I finish what I start*”), and sensation seeking (e.g., “*I’ll try anything once*”). Participants responded to items on a scale ranging from 1 (disagree strongly) to 4 (agree strongly). Higher subscale scores indicate greater impulsivity. In the current study, overall Cronbach’s alphas were 0.70 (0.83 for the urgency subscale, 0.85 for the lack of premeditation subscale, 0.76 for the lack of perseverance subscale, and 0.82 for the sensation seeking subscale).
Binge Eating Scale (BES)

Behavioral, cognitive, and emotional features of binge eating were assessed using the self-reported 16-item Binge Eating Scale [50,51]. For each question, participants were instructed to select, among statements, the one that best describes their behavior. For instance, items describe attitudes or behaviors from “*I feel capable to control my eating urges when I want to*” to “*Because I feel so helpless about controlling my eating, I have become very desperate about trying to get in control*” or from “*I rarely eat so much food that I feel uncomfortably stuffed afterwards*” to “*I eat so much food that I regularly feel quite uncomfortable after eating and sometimes a bit nauseous*”. Each statement is weighted either 0, 1, 2, or 3. Higher scores indicate greater binge eating symptoms (severity) and a score >17 is considered as an indicator of BED. Internal consistency in the current sample was excellent (Cronbach’s alpha = 0.83)

### 2.5. T1-Weighted MRI Acquisition and Voxel-Based Morphometry Measurements

T1-weighted three-dimensional (3D) turbo field echo images were acquired using a 3T whole-body MRI scanner (Philips, Ingenia, Philips Medical Systems) equipped with a 32-channel head coil at the Centre de recherche de l’IUCPQ. The following parameters were used: 176 sagittal 1.0 mm slices, repetition time/echo time (TR/TE) = 8.1/3.7 ms, field of view (FOV) = 240 × 240 mm^2^, and voxel size = 1 × 1 × 1 mm.

Grey matter density was assessed from each T1-weighted MRI using a standard voxel-based morphometry pipeline [28,52]. The preprocessing steps were the following: (1) image denoising [53]; (2) intensity non-uniformity correction [54]; and (3) image intensity normalization into range (0–100) using histogram matching. Images were then first linearly (using a nine-parameter rigid registration) and then nonlinearly registered to an average brain template (MNI ICBM152) as part of the ANIMAL software [55] and segmented into grey matter, white matter, and cerebrospinal fluid images. These steps remove global differences in the size and the shape of individual brains and transform individual grey matter density maps to the standardized MNI ICBM152 template space. Voxel-based morphometry analysis was performed using MNI MINC tools (http://www.bic.mni.mcgill.ca/ServicesSoftware/MINC (accessed on 25 January 2021) to generate grey matter density maps representing the local grey matter concentration per voxel. All image processing steps were visually quality controlled by an expert rater and all cases passed this visual quality check.

The average voxel-based morphometry grey matter densities were then calculated for each participant for six selected ROIs: (1) insula (combined regions 23 and 74 of CerebrA atlas [56]); (2) OFC (regions 7, 15, 58, and 66 of CerebrA atlas [56]); (3) caudal ACC (regions 30 and 81 of CerebrA atlas [56]); (4) rostral ACC (regions 8 and 59 of CerebrA atlas [56]); (5) vmPFC (combined regions 64, 65 and 88 of Glasser atlas [57]); and (6) DLPFC (regions 26, 67, 68, 70, 71, 73, 83, 84, 85, 86, 87, 97 and 98 of Glasser atlas [57]). For each ROI, the right and left hemisphere grey matter densities were pooled. These ROIs have been selected due to their potential implication in food reward, self-regulation, or emotional regulation processes and their previous associations with obesity and binge eating [10].

### 2.6. Statistical Analyses

Results were reported by mean and SD for continuous variables and number and percentage for dichotomous variables. Sex and group comparisons (i.e., based on VAI score) were performed using a *χ*^2^ test and an independent-sample t-test if applicable. Nonparametric tests were used when necessary (Mann–Whitney U test and Fisher’s exact test). We used raincloud plots [58] to show the distribution of VAI among men and women and Shapiro-Wilk tests were performed. High- or low-VAI groups were defined based on the mean VAI score, independently in men and women. Three participants were removed from the final analysis because of missing or outlying values. ROIs mean value for high- or low-VAI groups were compared independently according to sex using independent sample t-test and general linear model with age included as covariate.

Pearson correlations were performed between adiposity measurements and binge eating score. The strength of the relationship was qualified as weak (correlation coefficient r < 0.40), moderate (r = 0.40 to 0.69), or strong (r ≥ 0.70) [59].

Moderated mediation analysis models were performed to explore the possibility of direct and indirect relations between VAI and binge eating severity, using sex as potential moderator of the relation. Mediation models assessed whether covariance between two variables X and Y is explained by a third variable, the mediator M [60]. We tested whether grey matter density in ROIs significantly associated with VAI (as measured by the mean voxel-based morphometry value for the ROI _M_) mediated the relation between visceral adiposity (as measured by the VAI _X_) and binge eating severity (as measured by the BES score _ Y_). When relationships between visceral adiposity, grey matter density in ROI, and binge eating were different in men and in women, sex was used as a moderator in the model (_W_). Conservative bootstrap confidence interval (95%) based on 10,000 bootstrap samples was used to estimate significance, accounting for Type I error rate inflation.

All statistical analyses were conducted using SPSS software version 26.0.0.1 for Mac. Figures were generated using the Seaborn 0.10.0 python data visualization library and PtitPrince 0.1.5 package. The moderation–mediation analysis was performed using model 5 of the PROCESS macro version 3.4.1 [60] within SPSS version 26.0.0.1. *p* values <0.05 were considered as statistically significant and ROI-based analysis were Bonferroni corrected.

## 3. Results

### 3.1. Clinical Characteristics of Participants

Clinical characteristics of the study participants are shown in Appendix A. Mean age was 44.5 ± 8.7 years, and mean BMI was 43.6 ± 4.0 kg/m^2^. Only 18 of 79 participants had a diagnosis of type 2 diabetes. No significant difference was observed between men and women regarding clinical characteristics. Except for HDL-cholesterol that was lower in men, other biological parameters (lipid profile and glucose homeostasis) did not differ according to sex of the participants. Hip circumference, percentage of fat mass, and BFMI were higher in women compared to men (*p* ≤ 0.002 for all). Conversely, neck circumference was higher in men compared to women (*p* < 0.001). Overall mean VAI score was 2.43 ± 1.16 and no significant difference was observed between men and women (*p* = 0.455). Figure 1 shows that the distribution of VAI scores appear to be bi-modal in both men and women (Shapiro–Wilk test *p* values were 0.027 for men and 0.006 for women).

### 3.2. Correlations between Adiposity Measurements and Binge Eating Scores

In women, BES score was not correlated with BMI (r = −0.073; *p* = 0.589) but was negatively associated with percentage of fat mass (weak correlation, r = −0.308; *p* = 0.020, Figure 2). Moreover, BES score was significantly and positively correlated with waist-to-hip ratio (weak correlation, r = 0.264, *p* = 0.020) and VAI score (moderate correlation, r = 0.466; *p* < 0.001). In men, no significant correlation was found between BES score and BMI, waist-to-hip ratio, percentage of fat mass, or VAI score (Figure 2a,c,e,g).

In women, BES score was not correlated with BMI (r = −0.073; *p* = 0.589) but was negatively associated with percentage of fat mass (weak correlation, r = −0.308; *p* = 0.020, Figure 2). Moreover, BES score was significantly and positively correlated with waist-to-hip ratio (weak correlation, r = 0.264, *p* = 0.020) and VAI score (moderate correlation, r = 0.466; *p* < 0.001). In men, no significant correlation was found between BES score and BMI, waist-to-hip ratio, percentage of fat mass, or VAI score (Figure 2a,c,e,g).

### 3.3. Comparison of Biological and Psychological Parameters between Participants with High-VAI Versus Low-VAI in Women and Men Separately

Based on the distribution of VAI scores (Figure 1), women and men were subdivided in two subgroups according to the mean VAI score: high- versus low-VAI scores.

#### 3.3.1. Women

Among women classified in the high-VAI group, the VAI score was significantly higher than in the low-VAI group (*p* < 0.001, Table 1). Women with high-VAI had higher waist-to-hip ratio, neck circumference, and higher levels of triglycerides, glucose, insulin, HbA1c and a higher HOMA-IR index (*p* < 0.05 for all, Table 1) compared to those with low-VAI. Women with high-VAI also had lower concentrations of HDL cholesterol (*p* < 0.001). Women classified with a low-VAI had higher body fat mass as measured by % of fat mass and the body fat mass index compared to women with high-VAI (*p* ≤ 0.03 for all, Table 1). Table 2 shows the comparison of psychological parameters between the high- and low-VAI groups. Binge eating score was almost 4 points higher in women with high-VAI compared with those with low-VAI (*p* = 0.004). Depression and impulsivity related scores were not significantly different between groups.

#### 3.3.2. Men

Among men classified in the high-VAI group, the VAI score was significantly higher than in the low-VAI group (*p* < 0.001, Table 1). Men with high-VAI had higher triglyceride and lower HDL cholesterol concentration levels (*p* < 0.05 for all, Table 1) compared to men with low-VAI. None of the other anthropometric or biological parameters differed significantly between the groups (Table 1). Regarding psychological parameters, a significant difference was observed between groups for the sensation-seeking subscale of the UPPS-Impulsive Behavior Scale only, with a higher score in men classified in the high-VAI group (*p* = 0.036; Table 2).

### 3.4. Comparison of Voxel-Based Morphometry Grey Matter Density in Selected ROIs between Men and Women with High- Versus Low-VAI

Figure 3 shows mean voxel-based morphometry grey matter density for the selected ROIs in men and women with high- versus low-VAI (detailed data are available in Appendix A). The density of the caudal ACC was significantly higher in women with low-VAI (0.62 ± 0.08) compared to those with high-VAI (0.56 ± 0.08) after Bonferroni correction (adjusted *p* value = 0.042). The DLPFC tended to be higher in women with low-VAI, but this trend disappeared after Bonferroni correction. We found no significant difference between women with high- versus low-VAI for the other ROIs. In men, no significant difference was observed between groups. We found similar results when age was included as covariate (Appendix A).

### 3.5. Exploratory Mediation Analysis

Considering that the grey matter density of caudal ACC was significantly reduced in women with high-VAI (Figure 3), we next examined whether the association between VAI and BES score was mediated by the grey matter density of this brain region. Sex was used as a moderator for direct and indirect effects. The results of our mediation and moderation model are shown in Figure 4. The path from VAI to the BES score (direct effect) was positive and significant (β_c1_ = 2.375, s.e. = 0.701, *p* = 0.001), indicating that participants with higher VAI had a higher BES score. The moderation by sex was marginally significant (*p* = 0.0597), suggesting a possible difference between men and women regarding the association between VAI and BES score. More specifically, examination of the moderation showed that for women, the link between VAI and BES was positive and significant (conditional effect = 2.375, *p* = 0.001), whereas it was nonsignificant for men (conditional effect = −0.030, *p* = 0.978). The indirect effect of the VAI score on the BES score via the caudal ACC density was positive and significant (β_IE_ = 0.325; 95%CI = [0.019, 0.925]) and not moderated by sex. Overall, the model accounted for 25% of the variance in BES score.

## 4. Discussion

To the best of our knowledge, this is the first study to examine the associations between the VAI, binge eating behavior, and brain-related structure in men and women with severe obesity. Of our findings, two are of particular relevance: (1) the use of VAI (reflecting the metabolic alterations of obesity), which allowed us to identify relationships between eating behavior and brain-related regions that were not found when we used BMI, and (2) the relationship between adiposity, brain morphometry, and eating behavior that appeared to be sex-specific. Our results highlighted a moderate positive association between VAI and binge eating score in women only, while we found no significant association with BMI (*p* = 0.823 for men, *p* = 0.589 for women). More specifically, women with high-VAI were characterized by metabolic alterations, including dyslipidemia and insulin resistance, and by higher binge eating score and lower grey matter density in the caudal ACC compared to women with low-VAI. In men, no significant associations were found between visceral adiposity and binge eating severity and no significant differences were observed according to classification of high- versus low-VAI. The results in men can possibly be explained by the lack of power and should be interpreted with caution.

We found that women with high-VAI were characterized by higher neck circumference and waist-to-hip ratio as well as a less favorable metabolic profile, but lower fat mass percentage compared to low-VAI women. This inconsistency emphasizes that body composition tools currently used (e.g., bioimpedance) does not necessarily reflect the dysmetabolic nature of adipose tissue and should never be used alone. Women classified with high-VAI were also characterized by a higher binge eating score as opposed to women with low-VAI. Our results are consistent with previous results from a longitudinal study conducted by Berner et al. in which they found that women with a greater percentage of abdominal fat, measured by dual-energy X-ray (DXA) absorptiometry, were at higher risk for loss-of-control eating (i.e., uncontrolled eating), one of the key features of binge eating [61]. The authors also found that women with higher baseline trunk and abdominal fat percentage (DXA measurements) showed increases in loss-of-control eating episode frequency over a two-year follow-up, whereas it remained stable among women with lower percentage of fat in trunk and abdominal regions [61]. Succurro et al. found that BED participants with obesity had significantly lower HDL-cholesterol, and higher C-reactive protein, HOMA-IR and VAI score compared to non-BED participants with obesity [62]. Their results were statistically significant after adjusting for age, sex and BMI. While Leone et al. failed to observe significant associations between binge eating score and visceral or subcutaneous adipose tissue measured by ultrasonography, they found significant positive relationships between binge eating and waist circumference as well as negative relationships with body fat estimated by skinfold measurement [63]. Their results differ from our study mainly due to several methodological differences between studies. For instance, Leone et al. included men and women in their analyses despite evidence showing sex differences in abdominal body fat accumulation and eating behaviors [3,43,44]. Moreover, participants from our study were recruited based on the presence of severe obesity whereas participants from Leone et al.’s study covered a large spectrum of adiposity (normal weight to severe obesity). Therefore, our results suggest a positive association between adipose tissue dysfunction (i.e., metabolic alterations related to visceral adiposity) and binge eating severity in women. This association can be missed using BMI alone. Therefore, more comprehensive, and functional measurements of obesity should be considered in studies on eating behavior and in clinical practice.

Women with high-VAI were also characterized by lower grey matter density of the caudal ACC. This finding is consistent with our previous meta-analysis providing strong evidence that obesity is associated with lower grey matter density in brain regions involved in cognition and emotional regulation [28]. If the mechanisms underlying the link between obesity and reductions in grey matter volume measured by MRI remain largely unknown, recent evidence indicates that visceral obesity-related metabolic alterations, including insulin resistance, dyslipidemia, and low-grade chronic inflammation, might mediate the link between obesity and brain abnormalities [64,65]. Moreover, brain alterations observed in individuals with obesity may also involve chronic inflammation, increased oxidative stress or cellular autophagy associated with obesity [66]. In our study, women with high-VAI were characterized by dyslipidemia and higher insulin resistance as shown by higher HOMA-IR values. It has been suggested that insulin resistance may impact several insulin-sensitive brain circuitries involved in eating behavior such as prefrontal regions [39,40]. For instance, recent neuroanatomical studies with large sample size also showed that insulin resistance is associated with lower cortical thickness in frontoparietal and temporal brain regions [67,68]. Other studies also showed to the chronic, low-grade inflammation related to obesity as a potential mechanism explaining obesity-induced grey matter alterations [69]. Longitudinal studies with large sample sizes and a sex-specific design are needed to examine how these abdominal obesity-related metabolic abnormalities might impact the brain structure and function. 

Our exploratory moderation–mediation analysis also revealed that caudal ACC grey matter density could mediate the association between VAI and binge eating score. Neuroimaging studies demonstrated that several areas of the ACC play significant roles in cognition and emotion regulation [70]. The caudal or dorsal ACC seems to be involved in cognitive functions [70], but also in emotional reactivity [71,72], especially negative emotionality [73]. This is particularly relevant regarding the negative association we found between grey matter density in caudal ACC and the severity of the binge eating. As developed in the emotion regulation deficit theory of BED, negative emotion may be a trigger of binge eating behaviors [24] or uncontrolled eating [10]. Grey matter volume abnormalities in the ACC and medial OFC have also been reported in individuals suffering from BED [74]. Functional neuroimaging studies also provided evidence that individuals with BED have altered neural activity in ACC and medial OFC in response to high-calorie food images [75] as well as altered neural activity in the prefrontal control network (including the anterior medial OFC, vmPFC, medial PFC) in response to inhibitory control tasks [76,77]. These findings highlight the potential role of the caudal ACC in binge eating behavior, and this may involve negative emotionality. Moreover, as we found in our moderation–mediation analysis, potential sex differences may exist only for the direct association between VAI and BES and must be considered. 

Our study has strengths and several limitations to consider. Although our study has a good overall sample size, analyses were performed according to sex, which represent a strong aspect of our study, but also a limitation since it resulted in a relatively small sample size for men. The sample size also restrained the number of ROIs included in the analysis. Thus, other ROIs have not been included even if they appear to be involved in reward, self-regulation, and emotional regulation processes (i.e., the striatum and the hippocampus). The bariatric population normally includes around 75% of women, which may explain the smaller sample size in men. We failed to observe a significant association between visceral adiposity and binge eating score in men, but this result should be interpreted with caution. Similar studies should be replicated in men only with increased statistical power. Another limitation is the absence of accurate measurement of emotional eating or emotion dysregulation. In future studies, such measurements may help clarify the link between visceral adiposity, caudal ACC, and negative emotionality. Because BMI is not adequate to assess and explore the metabolic alterations associated with obesity, usual categorical approach may not be sufficient to assess eating behaviors. In future studies, the use of a comprehensive and systematized framework, such as the Research Domain Criteria (RDoC) matrix, should be considered to better understand eating behavior patterns [78,79]. The RDoC matrix is based on specific quantifiable dimensions of behavior (e.g., reward valuation or cognitive control) and corresponding units of analysis (from genes to behaviors and self-reports). Such dimensional approaches will help standardize eating behavior assessment and their relationship to adipose tissue dysfunction. Exploratory mediation analysis examining the potential causal relationship between visceral adiposity, binge eating severity, and brain structures should be conducted in men and women distinctively with larger sample sizes. Finally, pro-inflammatory cytokines known to be associated with visceral adipose tissue should be examined as they may be involved in mechanisms explaining the association between visceral adiposity and associated brain structure changes. 

## 5. Conclusions

In conclusion, visceral obesity-related metabolic alterations are associated with higher binge eating severity in women. This association is not observed in men. However, the small sample size in men prevent us to draw clear conclusions. Decreased grey matter density in the caudal part of the ACC, a region involved in cognition and emotion regulation, may influence this relationship but the causal relationship as well as the impact of visceral adiposity on emotional eating or emotional dysregulation needs to be confirmed.

## Figures and Tables

**Figure 1 brainsci-11-01158-f001:**
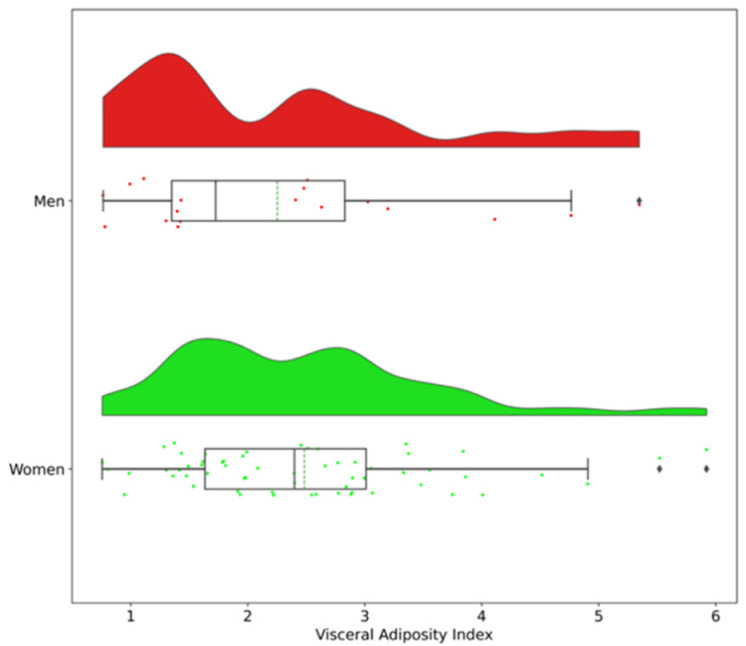
Representation of the distribution of visceral adiposity index (VAI) scores by sex using a raincloud plot. Median (solid line) and mean (dashed line) values are indicated in the boxplot.

**Figure 2 brainsci-11-01158-f002:**
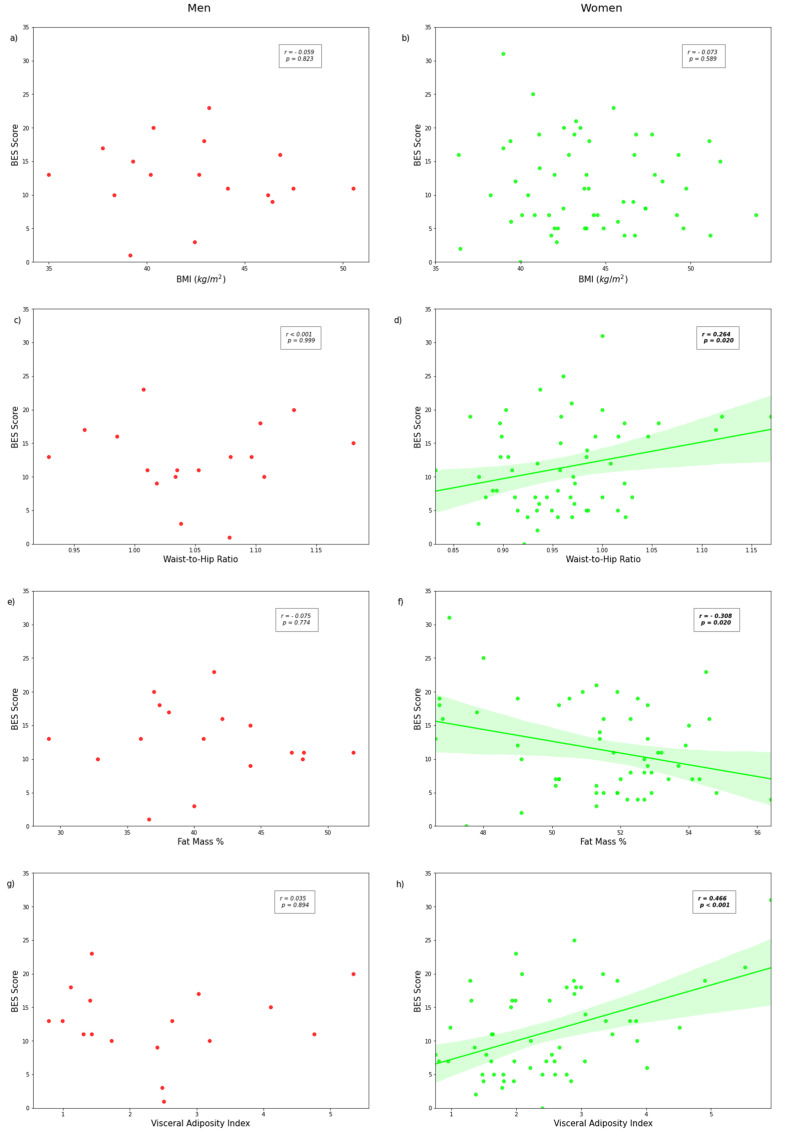
Correlations between binge eating scores and BMI ((**a**) for men and (**b**) for women), waist-to-hip ratio ((**c**) for men and (**d**) for women), fat mass % ((**e**) for men and (**f**) for women), or visceral adiposity index (VAI) ((**g**) for men and (**h**) for women). Figures show scatterplot with regression line and its 95% confidence interval in which relevant (using seaborn regplot). Pearson correlation coefficients and *p* values are shown. Significant associations are highlighted in bold.

**Figure 3 brainsci-11-01158-f003:**
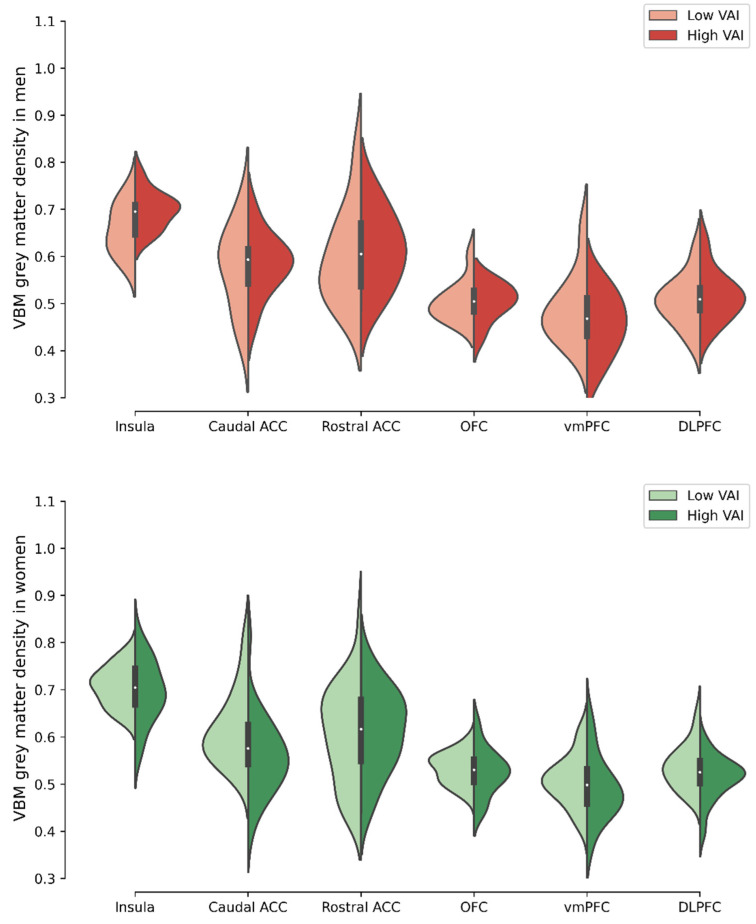
Voxel-based morphometry (VBM) value for our regions of interest in men and women with high- versus low- visceral adiposity index (VAI). Figures show violin plot. ACC: anterior cingulate cortex; OFC: orbitofrontal cortex; vmPFC: ventromedial prefrontal cortex; DLPFC: dorsolateral prefrontal cortex.

**Figure 4 brainsci-11-01158-f004:**
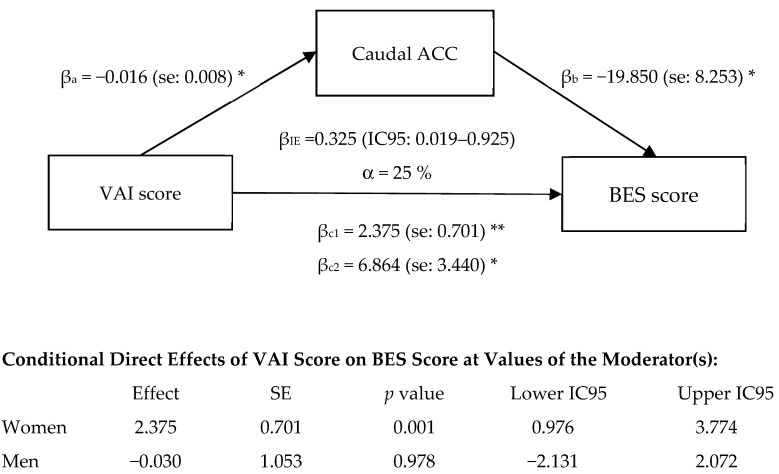
Voxel-based morphometry caudal anterior cingulate cortex (ACC) density as a mediator of the relationship between visceral adiposity index (VAI) and binge eating scale (BES) score, using sex as moderator. The figure shows unstandardized β regression coefficients with standard error (SE) or 95% confidence interval (IC95), with a number of bootstrap samples of 10,000. β_a_: effect of VAI score on caudal ACC; β_b_: effect of caudal ACC on BES score; β_c1:_ total effect of VAI on BES score; β_c2_: direct effect of sex on BES score; β_IE_: indirect effect of VAI on BES score; IC95: 95% confidence interval; α: percentage of the effect of VAI on BES score that is mediated by voxel-based morphometry caudal ACC density; se: standard error; * *p* < 0.05. ** *p* < 0.005.

**Table 1 brainsci-11-01158-t001:** Comparison of anthropometric and biological parameters between participants with high- versus low-visceral adiposity index (VAI).

	MEN			WOMEN		
	LOW-VAI	HIGH-VAI		LOW-VAI	HIGH-VAI	
	*n =* 10	*n =* 9		*n* = 32	*n* = 28	
	Mean ± SD	Mean ± SD	*p* value *	Mean ± SD	Mean ± SD	*p* value
Visceral Adiposity Index	1.2 ± 0.3	3.4 ± 1.1	<0.001	1.7 ± 0.4	3.4 ± 0.9	<0.001
**Anthropometric parameters**						
Age (yr)	50.8 ± 6.6	44.4 ± 7.7	0.113	44.0 ± 7.4	42.7 ± 10.2	0.569
BMI (kg/m^2^)	43.1 ± 4.9	40.9 ± 2.9	0.243	44.7 ± 4.0	43.4 ± 3.6	0.200
Waist circumference (cm)	134 ± 12	130 ± 8	0.356	128 ± 10	129 ± 8	0.713
Hip circumference (cm)	128 ± 12	122 ± 9	0.243	136 ± 9	131 ± 9	0.028
Neck circumference (cm)	46 ± 3	46 ± 2	0.780	39 ± 3	41 ± 2	0.033
Waist-to-hip ratio	1.0 ± 0.1	1.1 ± 0.1	0.661	0.9 ± 0.0	1.0 ± 0.1	0.004
Percentage of Fat Mass (%)	41.8 ± 6.3	39.5 ± 5.0	0.315	52.3 ± 2.2	50.4 ± 2.2	0.001
Body fat mass index (kg/m^2^)	18.3 ± 4.5	16.2 ± 3.0	0.156	23.4 ± 2.8	21.9 ± 2.5	0.031
**Biological parameters**						
Triglycerides (mmol/L)	1.0 ± 0.2	2.4 ± 0.8	<0.001	1.1 ± 0.3	1.8 ± 0.5	<0.001
Total cholesterol (mmol/L)	4.0 ± 1.2	4.6 ± 1.1	0.243	4.6 ± 0.7	4.5 ± 1.0	0.653
HDL cholesterol (mmol/L)	1.2 ± 0.2	1.0 ± 0.2	0.028	1.4 ± 0.3	1.1 ± 0.2	<0.001
LDL cholesterol (mmol/L)	2.3 ± 1.0	2.5 ± 1.0	0.780	2.6 ± 0.7	2.5 ± 0.8	0.676
Apolipoprotein B (g/L)	0.8 ± 0.3	1.0 ± 0.3	0.182	0.9 ± 0.2	1.0 ± 0.2	0.260
Fasting glucose (mmol/L)	6.5 ± 1.1	5.8 ± 0.5	0.315	5.8 ± 0.7	6.8 ± 2.4	0.032
Insulin (pmol/L)	185.2 ± 102.2	209.8 ± 73.5	0.549	137.2 ± 93.3	186.9 ± 96.9	0.048
HbA1c (%)	5.7 ± 1.1	5.5 ± 0.3	0.968	5.5 ± 0.5	6.2 ± 1.2	0.005
HOMA-IR index	9.0 ± 5.5	9.1 ± 3.4	0.999	6.0 ± 4.3	9.8 ± 6.7	0.013
TSH (mU/L)	2.3 ± 1.5	2.8 ± 1.3	0.400	2.6 ± 1.2	2.7 ± 1.4	0.651

VAI mean value for men: 2.25 and for women: 2.48; SD: standard deviation; BMI: body mass index; HDL: high-density lipoprotein; LDL: low-density lipoprotein; HOMA-IR: homeostatic model assessment for insulin resistance; HbA1c: hemoglobin A1c or glycated hemoglobin; TSH: thyroid stimulating hormone; * independent samples Mann–Whitney U Test.

**Table 2 brainsci-11-01158-t002:** Comparison of psychological parameters between participants with high- versus low-visceral adiposity index (VAI).

	MEN			WOMEN		
	LOW-VAI	HIGH-VAI		LOW-VAI	HIGH-VAI	
	*n* = 8	*n* = 9		*n* = 30	*n* = 27	
	Mean ± SD	Mean ± SD	*p* value *	Mean ± SD	Mean ± SD	*p* value
**Psychological parameters**						
Binge Eating Scale	14.4 ± 4.4	11.0 ± 6.2	0.277	9.1 ± 5.7	14.0 ± 6.6	0.004
Beck Depression Inventory	12.4 ± 6.8	8.3 ± 7.6	0.200	8.2 ± 7.3	11.1 ± 8.4	0.173
UPPS Impulsive Behavior Scale						
Negative urgency	28.0 ± 6.5	25.9 ± 5.0	0.743	24.8 ± 4.8	26.6 ± 5.7	0.202
Lack of premeditation	19.4 ± 3.1	20.6 ± 4.5	0.423	20.9 ± 3.7	21.8 ± 5.8	0.514
Lack of perseverance	18.6 ± 4.1	18.2 ± 4.8	0.888	17.6 ± 3.3	18.3 ± 4.5	0.549
Sensation seeking	27.5 ± 5.6	>33.7 ± 4.2	0.036	25.2 ± 6.6	24.3 ± 6.4	0.605

VAI mean value for men: 2.25 and for women: 2.48; SD: standard deviation; UPPS: urgency, premeditation (lack of), perseverance (lack of), sensation seeking; * independent samples Mann–Whitney U Test.

## Data Availability

The data presented in this study are not publicly available but are available on reasonable request from the corresponding authors.

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
