# Peer review of "Association between Visceral Adiposity Index, Binge Eating Behavior, and Grey Matter Density in Caudal Anterior Cingulate Cortex in Severe Obesity"

_brainsci, 2021, doi:10.3390/brainsci11091158_

Round 1

Reviewer 1 Report

The authors examined the associations between the VAI, binge eating behavior and volumetric correlates in men and women with severe obesity. First, the authors divided men and women into two groups based on a novel index – VAI score. Subsequently, the authors compared GMD of regions of interest (ROI) between low VAI and high VAI groups in men and women separately. Lastly, moderated mediation analysis models were performed. This is a very interesting study. However, I still have some questions that needs to be addressed.

Title

1) Gray matter density (GMD) is a measure often assumed to be highly related to volume, but it is not equivalent to GMV. The current study measured GMD, rather than GMV, and thus the title should be revised.

Abstract

1) Lines 27-28: The sentence is not clear - “Men and women were 27 divided into two groups according to mean VAI score (low or high).” It may need to be revised.

2) Psychological assessments of depression and impulsivity have not been mentioned in the abstract. And the related results have not been reported.

Introduction

1) What is the theory for choosing the six ROIs? Striatum is also a core structure involved in reward process, but why has not it been examined?

Materials and Methods

1) Do the authors consider including age as a covariate for the analyses?

2) Line 148: Is the visceral adiposity index typically computed using the formula in other studies? Are there any references for this?

Results

1) Line 310: The subtitle should not include “grey matter volume”, instead, it should be “density”.

2) It is interesting to find the low-VAI vs. high-VAI showed significant difference in sensation seeking of UPPS impulsivity in men only (even with a very sample size), but not women. Have the authors attempted to look at the relationships between sensation seeking, regional GMD, and BES?

Discussion

1) Lines 396-413: The authors have discussed the current findings and prior evidence that showed associations between GMV and obesity. How about the findings in previous studies regarding the relationships between GMD and obesity?

Conclusion

1) The authors haven emphasized the limitation of much smaller sample size in men. However, they may also need to be cautious when making the conclusion. In particular, the sentence “…This association is not observed in men …” needs to be revised. Likewise, a sentence “… No significant associations were found in men …” in the abstract needs to be revised based on the limitation of statistical power for the men group.

Reviewer 2 Report

This topic is interesting, but some points need a revision, please look at all these points:

  • Lines 68-74: Amygdala was never cited in the text. Why this author decision? Please look at this review: --- Neuroimaging in bulimia nervosa and binge eating disorder: a systematic review. J Eat Disord. 2018 Feb 20;6:3. doi: 10.1186/s40337-018-0187-1. 
  • Lines 356-358: "Indeed, our results highlighted a moderate posi357 tive association between VAI and binge eating score in women only, while we found no significant association with BMI" Does this represent a trend ? If yes, please insert the value.
  • Lines 56-62: "... hypertension, dyslipidemia, and type 2 diabetes. The increased cardiometabolic risk associated with disrupted eating behavior... " The role of dyslipidemia, diabetes and brain tumor should be reported in few lines here. Please consider this paper: Clinical Risk and Overall Survival in Patients with Diabetes Mellitus, Hyperglycemia and Glioblastoma Multiforme. A Review of the Current Literature. Int J Environ Res Public Health. 2020 Nov 17;17(22):8501. doi: 10.3390/ijerph17228501. 
  • Lines 363-364: "The results in men can possibly be explained by the lack of power and should be interpreted with caution" How do in their opinion authors explain those results?
  • Lines 400-403: "Even if the mechanisms underlying... might mediate the link between obesity and brain abnormalities " However pathomechanisms may involve chronic inflammation, increased oxidative stress or cellular autophagy associated with obesity.  ---Association between waist circumference and gray matter volume in 2344 individuals from two adult community-based samples. Neuroimage. 2015 Nov 15;122:149-57. 
  • Lines 429-431: "These findings highlight the... association between VAI and BES and must be considered." Also frontal eye field (FEF) seems to play a role in this complex system. Refs: --- A systematic review and meta-analysis of attentional bias toward food in individuals with overweight and obesity. Appetite. 2020 Aug 1;151:104710. doi: 10.1016/j.appet.2020    ----  Right Cortical and Axonal Structures Eliciting Ocular Deviation During Electrical Stimulation Mapping in Awake Patients. Brain Topogr. 2016 Jul;29(4):561-71. doi: 10.1007/s10548-016-0490-6. 
  • Statistical data appears correct.
